# SegPrompt: Using Segmentation Map as a Better Prompt to Finetune Deep Models for Kidney Stone Classification

**Wei Zhu**[1]    **Runtao Zhou**[1]    **Yuan Yao**[1]
**Timothy Campbell**[2]    **Rajat Jain**[2]    **Jiebo Luo**[1]
[1] **University of Rochester** [2] **University of Rochester Medical Center**

**Editors:** Accepted for publication at MIDL 2023

## Abstract

Recently, deep learning has produced encouraging results for kidney stone classification using endoscope images. However, the shortage of annotated training data poses a severe problem in improving the performance and generalization ability of the trained model. It is thus crucial to fully exploit the limited data at hand. In this paper, we propose SegPrompt to alleviate the data shortage problems by exploiting segmentation maps from two aspects. First, SegPrompt integrates segmentation maps to facilitate classification training so that the classification model is aware of the regions of interest. The proposed method allows the image and segmentation tokens to interact with each other to fully utilize the segmentation map information. Second, we use the segmentation maps as prompts to tune the pretrained deep model, resulting in much fewer trainable parameters than vanilla finetuning. We perform extensive experiments on the collected kidney stone dataset. The results show that SegPrompt can achieve an advantageous balance between the model fitting ability and the generalization ability, eventually leading to an effective model with limited training data.

**Keywords:** Kidney Stone Classification, Prompt Tuning, Parameter Efficient Finetuning

## 1. Introduction

Kidney stone disease (KSD) affects 10% of the US population during their lifetime and results in billions of dollars of the annual cost to society (Chewcharat and Curhan, 2021). The recurrent nature of KSD often causes multiple emergency room visits, hospital admissions, and surgical procedures for the patient. Over the last ten years, laser technology has evolved significantly (Kronenberg and Somani, 2018; Elhilali et al., 2017; Ibrahim et al., 2020). The most common procedure for KSD is ureteroscopy with laser lithotripsy (Heers and Turney, 2016). In this procedure, a semi-rigid or flexible 2-3mm ureteroscope is navigated into the urinary tract to the stone. It is then fragmented using a holmium:YAG laser. This type of laser has been the mainstay of KSD procedures for over 30 years (Zarrabi and Gross, 2011). In the past, it has been standard to fragment the stone into small pieces, which are then removed from the body using a small basket that is passed through the scope. This method allows the urologist to collect small pieces, which can then be sent to a laboratory for chemical analysis. However, this approach typically takes 1-2 months to get the classification result back to the physician, even though the patients can be in a critical condition and suffer from great pain  (Ochoa-Ruiz et al., 2022). In this paper, we focus on real-time stone-type prediction with the deep neural network (Ochoa-Ruiz et al., 2022).

Recently, deep learning-based methods have been developed to perform efficient diagnosis using endoscope images, and these methods often directly fine-tune the whole model (Ochoa-Ruiz et al., 2022; Estrade et al., 2022). However, similar to most medically related tasks (Zhu et al., 2020), the limited training data makes it hard to obtain a robust deep model that could generalize to unseen cases with vanilla finetuning (Zang et al., 2022). In this paper, inspired by the recent progress on Visual Prompt Tuning (VPT) (Jia et al., 2022), we propose SegPrompt for kidney stone classification by taking segmentation maps as prompts to tune the pretrained model. On the one hand, SegPrompt integrates the segmentation map into the training process to make the model aware of the regions of interest, which intuitively benefits the classification training process (Khan et al., 2019). The segmentation map is obtained with a pretrained Unet (Ronneberger et al., 2015). On the other hand, as a prompt tuning-based method, SegPrompt does not update the backbone model and thus has much fewer trainable parameters than finetuning. In this way, we can avoid the overfitting problem that often comes with small-scale training data. Moreover, our model allows the image and segmentation tokens to interact mutually with each other so that the model can make full use of the segmentation map.

We highlight our contributions as follows:

1. We propose SegPrompt, which regards segmentation maps as prompts to tune the pretrained deep model for kidney stone classification with limited training data.

2. SegPrompt incorporates the segmentation maps to facilitate the classification training process and only prompt-tunes a small part of the model, thus alleviating the overfitting problem and improving the classification performance.

3. We conduct thorough experiments on the collected dataset to validate the effectiveness of SegPrompt.

## 2. Related Work

### 2.1. Kidney Stone Classification

Both traditional machine learning and deep learning approaches have been used for kidney stone classification. Serrat *et al.* adopt random forest to classify kidney stone images with hand-crafted texture and color feature vectors (Serrat et al., 2017). Motivated by the encouraging results of deep medical image analysis (Ronneberger et al., 2015), Amado and Alejandro (Torrell Amado, 2018) exploit deep metric learning approaches such as Siamese Networks and Triplet Networks to learn the embedding for kidney stone images and use the k-nearest neighbor algorithm to classify testing images. Black *et al.* (Black et al., 2020) incorporate ex-vivo kidney stones images to finetune a pre-trained deep neural network and obtain reasonable results. Estrade *et al.* leverage transfer learning to classify mixed stones using surface and cross-section images of kidney stones (Estrade et al., 2022). In Manoj *et al.*'s work (Manoj et al., 2022), they present the visualization analysis of the well-trained kidney stone classifier with Grad-CAM. Finally, Ochoa-Ruiz *et al.* (Ochoa-Ruiz et al., 2022) use deep neural networks to classify kidney stones with in-vivo images from medical endoscopes. However, most of these methods finetune the whole model, potentially leading to an overfitting problem. In contrast, SegPrompt has fewer trainable parameters

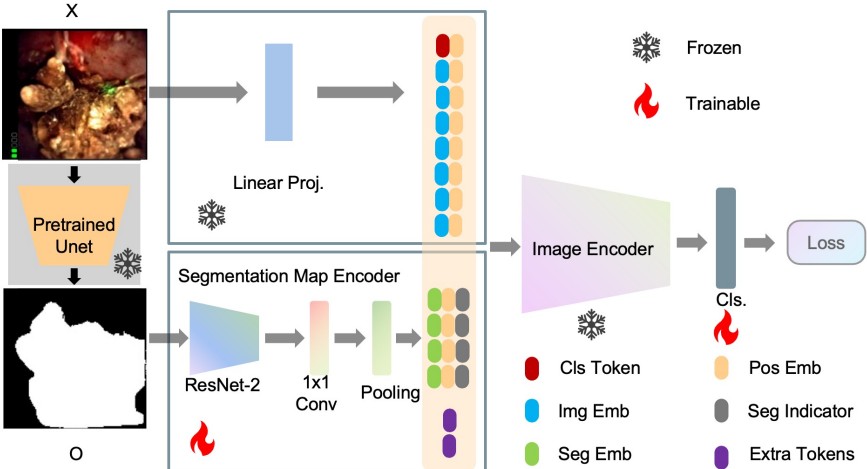

Figure 1: Block diagram of SegPrompt. We first extract the segmentation map with a pretrained Unet. The segmentation map is encoded into segmentation embeddings by the first two blocks of a pretrained ResNet18. We add the position embedding and segmentation indicator to the segmentation embedding to obtain segmentation tokens. Finally, we concatenate the image tokens, segmentation tokens, and extra learnable tokens and feed all tokens to the transformer backbone. We only update the segmentation map encoder and the last classifier during training.

to enhance the generalization ability and incorporates the segmentation map to facilitate the training.

## 2.2. Visual Prompt Tuning

Visual Prompt Tuning (VPT) was recently proposed to adjust a pretrained vision transformer for specific tasks with few trainable parameters and shows advantages in generalization ability over vanilla fine-tuning, particularly with limited training data (Jia et al., 2022). VPT simply adds learnable tokens to the input of vision transformers (Jia et al., 2022). Visual Prompting pads the original images with learnable pixels (Bahng et al., 2022). NOAH performs a neural architecture search to learn optimal prompt structure (Zhang et al., 2022). Unified vision and language prompt tuning are proposed to jointly tune the VL models (Zang et al., 2022). S-Prompt is proposed to handle domain incremental learning with prompt tuning (Wang et al., 2022). The prompts used by these methods are simply trainable parameters. In contrast, SegPrompt learns to generate prompts based on the segmentation map, and achieve a better balance between fitting and generalization ability.

## 3. Methodology

In this section, we present the proposed segmentation map-based prompt tuning framework for kidney stone classification. It is designed to improve the model's performance and

generalization ability by better exploiting the knowledge of segmentation maps with fewer trainable parameters.

## 3.1. Overview

Our method contains a frozen ViT backbone (Dosovitskiy et al., 2020), a segmentation map encoder, and a linear classifier. Similar to most medical image tasks, we also suffer from the scarcity of annotated kidney stone images, and it is expensive to collect more samples (Zhu et al., 2020). The shortage of training data makes it hard to obtain an effective model which could generalize to unseen cases. To alleviate the problem, on the one hand, we involve the segmentation map in the classification training so that the model is aware of the regions of interest. On the other hand, we tokenize the segmentation map into prompts to finetune the backbone model and only update the segmentation map encoder and the last linear classifier. Consequently, the much fewer trainable parameters empower our model with better generalization abilities to avoid the overfitting problem (Jia et al., 2022). Moreover, since we perform self-attention on both image and segmentation tokens (Vaswani et al., 2017), our method allows the model to exploit the knowledge of the segmentation map more comprehensively and flexibly. We show the block diagram of our method in Fig. (1).

## 3.2. Tokenize the Segmentation Map

We first show how to convert the segmentation map into prompt tokens with the proposed segmentation map encoder $h$. The encoder $h$ consists of the first two blocks of an ImageNet pre-trained ResNet18 followed by a projector, where the projector is composed of a 1x1 convolutional layer and an adaptive pooling layer (He et al., 2016). The convolutional layer of the projector is used to match the dimension of Resnet output to that of ViT while the pooling layer reduces the segmentation tokens to a desirable length. Moreover, the encoder $h$ also contains learnable tokens $P_s$, $Z_e$, and $r$, which will be introduced later. Given a training image $X$, we first obtain the binarized pixel-wise segmentation map $O \in \{-1, 1\}$ from a pretrained Unet (Ronneberger et al., 2015), where 1 denotes foreground regions with kidney stones, and $-1$ denotes background regions, the segmentation map embeddings $M = \{m^i\}_{i=1}^{l_m} \in R^{l_m \times d}$ can be obtained by flattening the output of segmentation map encoder as

$$M = flatten(h(O)), \tag{1}$$

where $d$ is the dimension of the backbone model. Then, to convert the embedding $M$ to tokens, we first add the learnable position embedding $P_s = \{p_s^i\}_{i=1}^{l_m} \in R^{l_m \times d}$ to retain the position information and then add a learnable indicator token $r \in R^d$ to enable the model to distinguish the segmentation tokens from the image tokens. Specifically, we obtain the segmentation tokens $Z_s = \{z_s^i\}_{i=1}^{l_m} \in R^{l_m \times d}$ by

$$z_s^i = m^i + p_s^i + r \ \ for \ \ i = 1, \ldots, l_m. \tag{2}$$

## 3.3. Prompt Tuning with Segmentation Tokens

To enable the model to better interact with and exploit the segmentation map, we propose to prompt-tune the model with the segmentation tokens (Jia et al., 2022). In particular, we

concatenate the classification token $z_{cls}$, image tokens $Z_x = \{z_x^i\}_{i=1}^l \in R^{l \times d}$, the segmentation tokens $Z_s$, and some extra learnable tokens $Z_e = \{z_e^i\}_{i=1}^{l_e} \in R^{l_e \times d}$ as input $Z$ to the transformer backbone

$$Z = [z_{cls}, z_x^0, \ldots, z_x^l, z_s^0, \ldots, z_x^{l_s}, z_e^0, \ldots, z_e^{l_e}] \tag{3}$$

The extra learnable tokens make the pretrained model better adapt to kidney stone classification. The classification token $z_{cls}$ and image tokens $Z_x$ are frozen during training (Dosovitskiy et al., 2020). We perform multi-head self-attention on the input tokens $Z$ and take the $z_{cls}$ from the last layer as the output, which will be further processed by the classifier to get the final prediction. We adopt cross-entropy loss to train the model. During training, we keep the transformer backbone frozen and only update the segmentation map encoder (including the corresponding position embedding $P_s$ and the indicator token $r$), the last classifier, and the introduced extra tokens $Z_e$.

We discuss several important features of SegPrompt as follows. Existing tuning methods strive to balance the generalization and fitting abilities to adapt a pretrained model for small-scale tasks. For example, finetuning suffers from the overfitting problem with overabundant learnable parameters, while VPT may underfit the target dataset, which deviates significantly from the pretrained dataset (Wortsman et al., 2022). Moreover, it is not trivial to integrate additional knowledge (e.g., segmentation map) into the training process for these methods. We empirically find that SegPrompt leads to a powerful model without severe overfitting and can effectively utilize extra knowledge. Specifically, compared with vanilla fine-tuning, the much fewer trainable parameters of SegPrompt significantly alleviate the overfitting problem. Compared with VPT, SegPrompt, equipped with the learnable segmentation map encoder, has a more powerful learning capacity and also does not severely distort the pretrained model because the number of newly introduced tokens (default set to 51) is much fewer than that of the original tokens (197 for ViT-B/16) (Dosovitskiy et al., 2020). To make use of additional knowledge, i.e., the segmentation map, SegPrompt allows image tokens and segmentation tokens to interact with and extract information from each other to improve classification performance. Compared with the joint classification and segmentation model, our framework is more flexible and can directly use human annotations when a good segmentation model cannot be obtained with the current data. Last but not least, one can simply extend SegPrompt to integrate other kinds of knowledge, such as patient demographic information (Daneshjou et al.), device/scanner information (Ji et al., 2022), text descriptions of the symptoms (Qin et al., 2022), and medical record. We leave these as future directions.

## 4. Experiments

### 4.1. Dataset

We collect 1496 kidney stone images from 5 different videos. We filter out low-quality and background images and obtain 867 images from 3 videos with COM (calcium oxalate monohydrate) stones and 629 images from 2 videos with CAP(calcium phosphate) stones. We split the dataset into training and validation sets in a video-wise fashion to prevent any data leaks. We perform 6-fold cross-validation, and the averaged accuracy, precision,

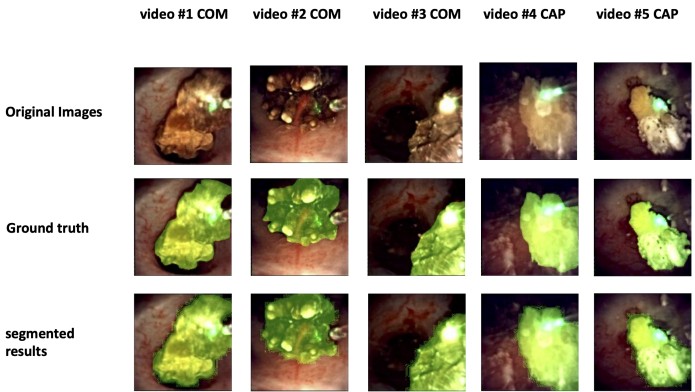

Figure 2: Illustration of segmentation results. Each column represents one image sample. The images come from the validation set of different folds.

recall, and F1 scores with standard deviation are reported on the validation sets composed of images from two hold-out videos.

### 4.2. Implementation of Segmentation Model

The kidney stone regions are labeled by an undergraduate student advised by a specialist, and we train a Unet (Ronneberger et al., 2015) implemented by the MMsegmentation framework (Contributors, 2020) to perform the segmentation. Besides the training data from different folds, we also leverage some extra images without kidney stone labels to facilitate the segmentation model training. In total, there are eight additional kidney stone videos without kidney stone labels, which provide 1860 additional images. We manually annotate the segmentation maps for these images and only include them in the segmentation model training. Finally, we obtain segmentation models with the pixel-level accuracy averaged over different folds as 96.7% and the dice score as 92.93%. The outputs are binarized to obtain final segmentation maps. We visualize segmentation results from the validation set in Fig. (2).

### 4.3. Baseline Methods and Implementation Details

Three finetuning-based models are included as the baselines to validate the superiority of the proposed SegPrompt for the kidney stone classification task as FT (Finetuning), FT-crop (Finetuning-crop), FT-concat (Finetuning-concat). FT uses raw images without segmentation maps. FT-crop takes the cropped regions of interest as input. As for FT-concat, we channel-wisely concatenate the segmentation maps to the images as the input. The concatenated images then passed through a 1x1 convolution layer to reduce their channel size to 3 before feeding them into the ViT backbone. We also implement FT-based ResNet for comparison. Moreover, we also compare our method with Visual Prompt Tuning (VPT) and VPT-Deep (Jia et al., 2022). VPT adds the learnable tokens to the input, while VPT-deep

Table 1: Kidney Stone Classification Results averaged over 6 Folds (corresponding to all possible combinations of videos). The best results are highlighted in bold. (%)

| Methods | Accuracy | Precision | Recall | F1 | AUC |
|---|---|---|---|---|---|
| FT | $95.07 \pm 3.2$ | $94.21 \pm 4.7$ | $95.46 \pm 2.7$ | $93.73 \pm 5.9$ | $95.37 \pm 3.4$ |
| FT-crop | $94.34 \pm 4.0$ | $94.14 \pm 4.2$ | $94.19 \pm 3.9$ | $92.96 \pm 5.6$ | $94.17 \pm 3.4$ |
| FT-concat | $95.18 \pm 2.5$ | $94.91 \pm 2.5$ | $95.13 \pm 2.3$ | $94.53 \pm 3.0$ | $95.10 \pm 2.9$ |
| ResNet50 | $94.75 \pm 2.7$ | $94.06 \pm 4.1$ | $95.49 \pm 2.0$ | $93.55 \pm 5.1$ | $95.18 \pm 1.6$ |
| ResNet50-crop | $95.28 \pm 3.8$ | $95.71 \pm 3.3$ | $94.66 \pm 3.7$ | $94.32 \pm 4.4$ | $94.58 \pm 3.7$ |
| ResNet50-concat | $96.72 \pm 2.6$ | $96.44 \pm 3.3$ | $96.93 \pm 2.5$ | $96.44 \pm 2.8$ | $96.84 \pm 2.5$ |
| VPT | $96.87 \pm 1.1$ | $96.60 \pm 1.7$ | $96.65 \pm 1.3$ | $96.07 \pm 2.8$ | $96.52 \pm 1.8$ |
| VPT-Deep | $95.85 \pm 0.8$ | $95.49 \pm 1.2$ | $95.60 \pm 1.2$ | $95.13 \pm 2.4$ | $95.32 \pm 1.7$ |
| SegPrompt | $\mathbf{99.56 \pm 0.3}$ | $\mathbf{99.45 \pm 0.4}$ | $\mathbf{99.60 \pm 0.3}$ | $\mathbf{99.45 \pm 0.5}$ | $\mathbf{99.57 \pm 0.4}$ |
| SegPrompt-Deep | $99.19 \pm 0.3$ | $99.06 \pm 0.3$ | $99.24 \pm 0.3$ | $99.26 \pm 0.2$ | $99.23 \pm 0.5$ |

adds different tokens for each layer. Similar to VPT and VPT-deep, we also implement two variants of SegPrompt as SegPrompt and SegPrompt-Deep.

For all methods, the standard image preprocessing steps, such as resizing and normalization, have been applied to the training images. The batch size, number of epochs, and learning rate are 16, 20, and 0.001, respectively. We adopt an ImageNet pretrained ViT-B/16 as the backbone, which contains 196 image tokens and one classification token (Dosovitskiy et al., 2020). As for the proposed SegPrompt (Deep), we set the number of segmentation tokens to $l_s = 49$, and the extra learnable tokens to $l_e = 2$. The number of learnable tokens for VPT (Deep) is searched from $\{8, 16, 32, 51\}$.

### 4.4. Experimental Results

We present the experimental results in Table 1, and draw several interesting conclusions according to the results. First, FT outperforms FT-crop, suggesting that the cropped clean kidney stone images do not benefit the classification performance, which in turn suggests that the surrounding background regions contain critical information. Second, FT-concat utilizes the segmentation map without removing the background regions and it slightly outperforms FT for 0.8% in terms of F1 score. This shows that the classification performance could be boosted by properly exploiting the information of the segmentation map. The performance of FT-ResNet has consistent results. Third, the overfitting problem is crucial in adapting the pre-trained models to the kidney stone classification with limited training data. VPT and VPT-deep outperform all FT-based methods with much fewer trainable parameters (Jia et al., 2022). In particular, VPT obtains a 1.54% improvement over FT-concat in terms of the F1 score and also surpasses its direct counterpart VPT-deep which has much more trainable parameters. Finally, the proposed SegPrompt makes better use of the segmentation map with few trainable parameters and achieves the best overall performance. For example, SegPrompt improves the F1 score from 96.07% to 99.45% compared with the second-best method VPT. We also note that SegPrompt-deep only slightly degrades the performance compared with SegPrompt, which shows the possibility of applying our method to handle large-scale tasks that require more learnable parameters to fit the training set.

Table 2: Ablation Studies. (%)

| Methods | Accuracy | Precision | Recall | F1 | AUC |
|---|---|---|---|---|---|
| SegPrompt | $99.56 \pm 0.3$ | $99.45 \pm 0.4$ | $99.60 \pm 0.3$ | $99.45 \pm 0.5$ | $99.57 \pm 0.4$ |
| SegPrompt w/o $r$ | $99.07 \pm 0.6$ | $98.90 \pm 0.7$ | $99.11 \pm 0.5$ | $98.87 \pm 0.8$ | $99.08 \pm 0.4$ |
| SegPrompt w/o $z_e$ | $99.38 \pm 0.5$ | $99.34 \pm 0.5$ | $99.36 \pm 0.8$ | $99.30 \pm 0.3$ | $99.42 \pm 0.5$ |

Table 3: Performance with the different number of segmentation tokens. (%)

| # $l_s$ | Accuracy | Precision | Recall | F1 | AUC |
|---|---|---|---|---|---|
| 25 | $98.81 \pm 1.0$ | $98.75 \pm 0.9$ | $98.73 \pm 1.0$ | $98.69 \pm 0.9$ | $98.70 \pm 0.8$ |
| 36 | $99.13 \pm 0.6$ | $99.16 \pm 0.4$ | $98.87 \pm 1.2$ | $98.78 \pm 1.4$ | $98.92 \pm 0.6$ |
| 49 | $\mathbf{99.56 \pm 0.3}$ | $\mathbf{99.45 \pm 0.4}$ | $\mathbf{99.60 \pm 0.3}$ | $\mathbf{99.45 \pm 0.5}$ | $\mathbf{99.57 \pm 0.4}$ |
| 64 | $99.28 \pm 0.2$ | $99.32 \pm 0.1$ | $99.22 \pm 0.3$ | $99.18 \pm 0.4$ | $99.24 \pm 0.3$ |
| 81 | $99.22 \pm 0.6$ | $99.14 \pm 0.7$ | $99.25 \pm 0.6$ | $99.13 \pm 0.7$ | $99.23 \pm 0.8$ |

## 4.5. Ablation Studies

Extensive ablation studies have been performed to verify the effectiveness of different components of SegPrompts. All variants are developed based on SegPrompt instead of SegPrompt-Deep due to its superior performance and simplicity. We first study the importance of indicator token $r$ and the extra learnable tokens $z_e$. The indicator token $r$ enables the model to distinguish the segmentation map tokens from other tokens, and the extra learnable tokens could make the pretrained model better adapt to our task. The two variants are denoted as SegPrompt w/o $r$ and SegPrompt w/o $z_e$, respectively. The experimental results are shown in Table 2. According to the results, we find that the indicator token is essential for SegPrompt while the extra learnable tokens further slightly improve the performance. We also conduct experiments to study the influence of different numbers of segmentation tokens $l_s$, and select $l_s \in \{25, 26, 49, 64, 81\}$. Based on the results shown in Table 3, the increasing number of segmentation tokens benefits the final performance, while over-large values may lead to a slight decrease. We default set $l_s = 49$.

## 5. Conclusions and Future Work

In this paper, we present a novel segmentation map-based prompt tuning method for kidney stone classification with limited data, named SegPrompt. We first employ a well-trained Unet to extract the segmentation maps, which are further converted to segmentation tokens by a segmentation map encoder. Then SegPrompt takes the concatenation of image, segmentation, and some extra tokens as input to the transformer. During training, we only prompt-tune the segmentation map encoder and the linear classifier. SegPrompt can better exploit the knowledge of segmentation maps with few trainable parameters and significantly outperforms existing methods for kidney stone classification. The main limitation of our work is the small-scale training dataset, and we will collect more data to improve our model for more types of kidney stones. Moreover, we plan to extend our work to other medical tasks to further validate the effectiveness of SegPrompt (Daneshjou et al.; Qin et al., 2022).
**Acknwledgments** This work is supported in part by NSF #2050842, NIH 1P50NS108676-01, and NIH 1R21DE030251-01.

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
