# OpenReview forum: "SegPrompt: Using Segmentation Map as a Better Prompt to Finetune Deep Models for Kidney Stone Classification"
_MIDL.io/2023/Conference — MIDL 2023 Poster_

### Official Review · Reviewer_CYQw · 2023-02-04

**Confidence:** 3
**Preliminary Rating:** 3
**Recommendation:** Poster

**Summary:**

This paper proposes a deep learning framework for the classification of kidney stones subtypes using segmentation maps to help the subsequent classification of kidney stones.
The proposed method is evaluated on a quality filtered selection of images from laparoscopic surgery videos in terms of accuracy, F1 score, Precision and Recall

**Strengths:**

The experiments appear overall well designed with care for cross-validation and indication of performance variability
The results are carefully interpreted with insight on the meaning of the findings and interesting points made


**Weaknesses:**

Motivation is not really clear as the reader only learns at the setting of the experiments what are the two subtypes of kidney stones to classify but there is no explanation as to why these two subtypes need to be separated.
There are some statements of outperformance but these are not supported by any statistical test. Having a ROC AUC for the classification performance would have been useful
The authors state the relevance of the proposed method when learning with fewer examples but there are no experiments to support that aspect of the paper.

**Deanonymize Review:**

no

**Detailed Comments:**

There is a general slight lack of clarity in the paper probably due to the ordering in which the information is sometimes provided. It is not clear for instance how the additional images are properly "leveraged" - Do they require kidney segmentation if not the full annotation?

**Paper Type:**

methodological development

**Questions To Address In The Rebuttal:**

The clinical rational should be made clearer from the start especially regarding the different types of stones considered and the reason for the need of classification
The relevance in terms of training amount should be better justified

---

### Official Review · Reviewer_BE5B · 2023-02-06

**Confidence:** 3
**Preliminary Rating:** 4
**Recommendation:** Poster

**Summary:**

This paper seems to leverage developments in transformers to kidney stone classification, complementing tag information with segmentation maps of the stones.

The method is evaluated on a private small dataset of around 1500 images.

Due to my limited knowledge on the topic of transformers, for now I am sticking to a conservative borderline (the results seem simply _bizarre_ to me at the moment, I hope this is clarified during rebuttal). But I intent to close the gap in my knowledge during the rebuttal.

**Strengths:**

- Seemingly very good results
- Seemingly solve the problem at hand (no further research required, the trained model can be deployed in a clinical setting)
- The method seems to be quite data efficient, leveraging a pre-trained model to fine-tune to the task

**Weaknesses:**

- The paper is rather difficult to follow if one is not up to date with the transformers litterature
- The reporting of the results is not clear, results by fold would be useful, and clarifying if the authors are reporting train or validation peformances would help

**Deanonymize Review:**

no

**Detailed Comments:**

Misc:
- Page 3: "Iy is designed toimprove" -> "It is designed to improve"
- \text should be used to format words in equations (such as $z_s^i = m^i + p^i_s + r \text{ for } i=1,...,l_m$ in Eq. (2).

**Paper Type:**

validation/application paper

**Questions To Address In The Rebuttal:**

It seems that the rebuttal period last one week only, so it would be great if the authors could respond before the end of it, so we can actively engage in a conversation (point 5 in reviewer guidelines https://2023.midl.io/reviewer-guidelines.html )

### On the seemingly perfect results

While I do not know much about transformers at the moment, I remain _very_ surprised by the results (99.5+% on all 4 metrics), which could be explained by either: 1) a very trivial classification problem 2) the sheer awesomeness of transformers, in which case I should probably drop my current research project.

To get a point of comparison, how would a standard CNN (with and without the segmentation map added as extra channel), trained from scratch, perform? 1500 images, with proper data augmentation and regularization techniques, can be enough for very constrained problems such as this one. Right now I am simply puzzled, as the authors do not seem to notice that the problem is basically solved.


---
### Other
- Maybe I missed the information, but how many classes are there in this task?

- The authors claim that the base methods they compare to suffer from "over-fitting", yet they achieve a stunning 95% accuracy/precision/recall/f1. Is that really supposed to demonstrate a massive over-fitting?


- The authors, in the related works, cite recent works that seem very relevant to the task at hand:
> Black et al. (Black et al., 2020) incorporate ex-vivo kidney stones images to finetune a pre-trained deep neural network and obtain reasonable results. Estrade et al. leverage transfer learning to classify mixed stones using surface and cross-section images of kidney stones (Estrade et al., 2022). In Manoj et al.’s work (Manoj et al., 2022), they present the visualization analysis of the well- trained kidney stone classifier with Grad-CAM. Finally, Ochoa-Ruiz et al. (Ochoa-Ruiz et al., 2022) use deep neural networks to classify kidney stones with in-vivo images from medical endoscopes. However, most of these methods finetune the whole model, potentially leading to an overfitting problem.

Why not comparing to them? This would also demonstrate the main claim of the authors (less parameters leading to less overfitting, and I reiterate: no overfitting is demonstrated at the moment).

---
**Final rating:** I thank the authors for their response, and their additional baseline experiment on ResNet50. This clarifies a lot to me (I was wondering if some training and testing data hadn't been mixed). I am increasing my score from `borderline` to `weak accept`, and confidence from `2` to `3`.

---

### Official Review · Reviewer_twWC · 2023-02-09

**Confidence:** 3
**Preliminary Rating:** 3
**Recommendation:** Poster

**Summary:**

This paper presents SegPrompt for addressing the task of kidney stone classification using endoscope images. The network relies on a pre-trained U-Net to produce segmentation maps, and leverage the segmentation maps as prompts to improve classification performance. The method is evaluated on a collected dataset, and results look promising.

**Strengths:**

The paper is well written, and it's easy for reviewer to get the idea and technical details of the method. The built dataset can be a valuable resource to the community if released. The idea is clear and interesting. The performance of the method is promising in comparison with the competitors.

**Weaknesses:**

While the idea is interesting, there are several issues regarding the method and experiments.

First, one clear downside of the proposed approach is that it heavily relies on segmentation annotations. This may limits the usage of the method in clinical practise, and actually will prevent the extension of the method to other medical tasks, as authors mention in the conclusion part. In addition, it is not clear how are the segmentation annotations collected.

Second, some experimental details should be clarified. Regarding the training/validation sets, what are the numbers of training / validation samples? In addition, I am actually thinking which set are the experiments results obtained from. Is it from the validation set or actually from a leave-out testing set?

Third, regarding the pipeline of the method, I am curious how about to directly train a segmentation network for joint segmentation and classification. What's the advantage of the proposed two-stage method agains such a simplified one-stage baseline.

**Deanonymize Review:**

no

**Paper Type:**

methodological development

**Questions To Address In The Rebuttal:**

Please refer to the issues mentioned in the weaknesses part. Particularly, more discussions and results on the method weaknesses, experimental details, and comparison with baseline models should be provided in the rebuttal.

---

### Meta-Review · Area_Chair_w2Ti · 2023-02-24

**Recommendation:** Accept (Poster)
**Confidence:** 3

**Metareview:**

The authors present a DL method for classifying kidney stones subtypes. They employ segmentation maps as additional prompts.

Overall, to me this paper remain quite borderline. Overall, the concerns of the reviewers seem to revolve around clarify of the paper and motivation, and the experimental setup (e.g. lack of certain baselines, some confusion over the over-fitting comments, etc). Upon the rebuttal, it seems like the problem remains narrow, and while the authors properly justify not being able to run certain baselines due to missing code/data, the insights we are left to take form the paper are narrow (specific task, specific domain, requiring manual segmentations for a classification problem, etc). Nevertheless, for this particular task, the paper warrants some interesting discussion.

I am a tad confused about the discussion around transformers (with reviewer CYQw), as the authors seem to defer their choice of transformers to previous literature, e.g. with comments like "transformers have been proven effective for many image recognition tasks.". This is very hand-wavy -- transformers are complicated machines that have only been shown to be superior in other very specific domains. Even in general computer vision, where transformers are certainly 'hot/popular' literature, most careful analysis show that CNNs are as good (e.g. see convnext paper). In medical imaging, I am not familiar with literature that has actually shown them to be superior, although they are being often examined. I have no concerns with authors using transformers, but the justification and answer to reviewer seem insufficient.

Overall, I would say the contribution is narrow but perhaps offers some interesting topics of discussion. I recommend a poster acceptance.